# Detection of *Pneumocystis jirovecii* in Patients with Severe COVID-19: Diagnostic and Therapeutic Challenges

**DOI:** 10.3390/jof7080585

**Published:** 2021-07-22

**Authors:** Victor Gerber, Yvon Ruch, Thiên-Nga Chamaraux-Tran, Walid Oulehri, Francis Schneider, Veronique Lindner, Valentin Greigert, Julie Denis, Julie Brunet, François Danion

**Affiliations:** 1CHU de Strasbourg, Service de Maladies Infectieuses et Tropicales, Fédération de Médecine Translationnelle de Strasbourg, Université de Strasbourg, 67000 Strasbourg, France; victor_gerber@hotmail.fr (V.G.); yvon.ruch@chru-strasbourg.fr (Y.R.); 2Service d’Anesthésie-Réanimation Chirurgicale, Hôpital de Hautepierre, Hôpitaux Universitaires de Strasbourg, Avenue Molière, CEDEX, 67098 Strasbourg, France; thiennga.chamaraux-tran@chru-strasbourg.fr; 3Institut de Génétique et de Biologie Moléculaire et Cellulaire (IGBMC), CNRS UMR7104, INSERM U1258, Université de Strasbourg, 67400 Illkirch, France; 4Service d’Anesthésie-Réanimation, Nouvel Hôpital Civil, Hôpitaux Universitaires de Strasbourg, 67000 Strasbourg, France; walid.oulehri@chru-strasbourg.fr; 5Service de Médecine Intensive Réanimation, Hautepierre, Hôpitaux universitaires de Strasbourg, 67000 Strasbourg, France; francis.schneider@chru-strasbourg.fr; 6Département de Pathologie, Hôpitaux Universitaires de Strasbourg, Hôpital de Hautepierre, Avenue Molière, CEDEX, 67098 Strasbourg, France; veronique.lindner@chru-strasbourg.fr; 7Department of Molecular Microbiology, Washington University School of Medicine, St. Louis, MO 63130, USA; Valentin.greigert@gmail.com; 8Laboratoire de Parasitologie et de Mycologie Médicale, Plateau Technique de Microbiologie, Hôpitaux Universitaires de Strasbourg, 67000 Strasbourg, France; julie.denis@chru-strasbourg.fr (J.D.); julie.brunet@chru-strasbourg.fr (J.B.)

**Keywords:** pneumocystosis, *Pneumocystis jirovecii*, COVID-19, SARS-CoV-2, fungi

## Abstract

Cases of *Pneumocystis jirovecii* pneumonia (PCP) in patients suffering from COVID-19 were described in patients with various comorbidities and outcomes. The diagnosis of PCP in these patients is difficult due to clinical and radiological similarities. We carried out this study in order to better describe potentially at-risk patients and their outcomes. We retrospectively analyzed all patients with a *P. jirovecii* PCR performed in bronchoalveolar lavage fluid, tracheal aspirate, or sputum within a month after the COVID-19 diagnosis. Fifty-seven patients with COVID-19 infection were tested for *P. jirovecii*. Among 57 patients with COVID-19, four patients had a concomitant positive *P. jirovecii* PCR. These four patients were elderly with a mean age of 78. Two patients were immunocompromised, and the two others presented only diabetes mellitus. Three patients presented an ARDS requiring transfer to the ICU and mechanical ventilation. All patients presented lymphocytopenia. Three patients had probable PCP, and one had proven PCP. All patients died within two months after hospital admission. These co-infections are rare but severe, therefore, PCP should be considered in case of worsening of the condition of patients with severe COVID-19.

## 1. Introduction

COVID-19 causes a wide spectrum of symptoms ranging from asymptomatic disease to life-threatening severe acute respiratory distress syndrome (ARDS) and death [1]. This disease can be complicated with fungal superinfection, particularly aspergillosis [2]. Cases of *Pneumocystis* pneumonia (PCP) were reported in HIV-positive subjects and kidney transplant recipients as well as immunocompetent patients suffering from COVID-19 [3,4,5]. A study found that two among 145 patients (1.4%) with severe COVID-19 had a positive polymerase chain reaction (PCR) for *Pneumocystis jirovecii* [6]. Another one found that a *P. jirovecii* PCR was positive in 10 of 108 (9.3%) patients with severe SARS-CoV-2 infection [7]. The diagnosis of PCP during COVID-19 is challenging, and the need to treat patients with a positive *P. jirovecii* PCR in this context remains unclear. We analyzed all consecutive *P. jirovecii* PCR performed in COVID-19 patients in our center and described four cases of PCP, including one proven case and three probable ones following EORTC/MSG criteria [8].

## 2. Materials and Methods

Among adult patients with confirmed SARS-CoV-2 infection between 1 March 2020 and 1 February 2021, hospitalized in Strasbourg University Hospital, we retrospectively analyzed all patients with a *P. jirovecii* PCR performed in bronchoalveolar lavage (BAL) fluid, tracheal aspirate, or sputum. During the ongoing COVID-19 pandemic, no microscopic diagnosis was done.

All COVID-19 patients with a positive *P. jirovecii* PCR in a sample harvested within a month after the COVID-19 diagnosis were included. Patients who were not followed in our center were excluded. The medical records were analyzed to collect the medical history as well as demographic, clinical, biological, radiological, and microbiological data.

The study was approved by the Ethics Committee of the University Hospital of Strasbourg (No. CE-2020-51). Oral non-opposition to participation to the study was sought, and the patients who expressed opposition to participate were not included.

## 3. Results

Among patients with SARS-CoV-2 infection during the inclusion period, 57 had a concomitant investigation for *P. jirovecii*. Four patients (7.1%) had a positive *P. jirovecii* PCR and were diagnosed with PCP. Patients’ demographic data and underlying diseases are described in Table 1; microbiological, biological, and clinical data are described in Table 1. 

The patients were elderly, with a mean age of 78 (range 70–83). Two patients were immunocompromised and at risk of PCP before the occurrence of COVID-19, one with rheumatoid arthritis treated by steroid therapy and one with history of hematopoietic stem cell transplantation and with an Ewing’s sarcoma metastasized to the lung and treated by cyclophosphamide (Sandoz, Levallois-Perret, France) and doxorubicine (Accord-healthcare, Lille, France). The other two patients only had diabetes mellitus. In addition, three patients were treated with systemic steroids for COVID-19. All these patients presented with lymphocytopenia at admission.

Three of four patients presented an ARDS requiring a stay in the ICU and mechanical ventilation. All of them had significant CT scan lesions with ground-glass opacities affecting more than 50% of the lung. *P. jirovecii* isolation was performed by PCR on tracheal aspirate for two patients and on BAL for the others. The BAL cytology showed round microorganisms suggestive of *Pneumocystis* cysts in one patient (Figure 1). Serum (1,3)-β-D-glucan (Fungitell cape Cod, East Falmouth, MA, USA) performed in two patients was positive.

All patients were treated by trimethoprim-sulfamethoxazole (TMP-SMX, Roche, Boulogne-Billancourt, France) for PCP. For one of them, TMP-SMX was stopped quickly after a skin rash and switched with atovaquone. All the patients also received broad-spectrum antimicrobial therapies for bacterial or viral superinfection.

All four patients died within two months (range 3–50 days) after hospital admission. Three of them died in the ICU. The last patient (patient 3) died suddenly four days after TMP-SMX introduction for proven PCP in rehabilitation service; his death was possibly attributed to PCP. The diagnosis was made by positive *P. jirovecii* PCR and cytology from BAL (Figure 1) after pulmonary worsening 40 days after hospital admission.

## 4. Discussion

Here, we identified four cases of PCP among 57 COVID-19 patients screened for *P. jirovecii* using PCR in BAL, tracheal aspirate, or sputum. Three were classified as probable and one as proven following EORTC/MSG criteria due to a positive BAL cytology [8].

Regarding risk factors, two patients had previous immunocompromised condition and three received steroids for COVID-19 therapy. None of the patients were treated with any other anti-inflammatory drug such as tocilizumab. The prognosis was poor for all patients.

The prevalence of positive *P. jirovecii* PCR and PCP in COVID-19 patients is unknown. Blaize et al. found positive *P. jirovecii* PCR in two of 145 (1.4%) SARS-CoV-2 infected patients in ICU [6]. Another study found 9.3% (10/108) of positive *P. jirovecii* PCR in COVID-19 patients presenting with ARDS [7].

Several cases of co-infections of SARS-CoV-2 with *P. jirovecii* were also described in patients with known risk factors for PCP. A case was described in a newly diagnosed HIV patient with COVID-19 who presented a severe depletion of CD4 T cells, fine reticular changes in CT scan, and an elevated level of lactate dehydrogenase (LDH), successfully treated with TMP-SMX [3]. Another case involved a renal transplant recipient with lymphocytopenia treated with tacrolimus, mycophenolate mofetil, and methylprednisolone [4,9]. He died despite a treatment by TMP-SMX. Interestingly, a case in an 84-year-old patient without history of immunosuppression was described [5]. She presented with severe COVID-19 and was hospitalized in the ICU. The diagnosis was made on lymphocytopenia, elevated serum (1,3)-*β*-D-glucan level, positive *P. jirovecii* PCR on a tracheal aspirate, cystic changes on the CT scan, and after a favorable outcome under treatment by TMP-SMX [5]. In severe COVID-19 cases, CD4+ T and CD8+ T cell levels could be remarkably low [1]. This lymphocytopenia was suggested as a risk factor for PCP, but the study by Blaize et al. did not find any link between the lymphocytopenia encountered in severe COVID-19 cases and the occurrence of PCP [6].

The diagnosis of PCP in COVID-19 patients and the distinction between infection and colonization are very challenging [6]. Patients hospitalized in the ICU for COVID-19 might be at risk of PCP due to mechanical ventilation, use of corticosteroid therapy, or the existence of a cytokine storm leading to marked alveolar-interstitial pulmonary tissue [1]. However, there are radiological similarities between the two infections, the presence of cysts or fine reticular changes on CT scan being in favor of pneumocystosis, but similarities are not constant and are sometimes difficult to see [3,5,10]. The high sensitivity of PCR can lead to overdiagnosis of *P. jirovecii* infection in colonized patients, and the distinction between colonization and PCP can be difficult, especially in immunocompetent patients [5,6,7,10]. Direct examination is usually not performed in COVID-19 patients. The use of a staining methodology (Giemsa, Gomori-methenamine-silver stain, toluidine blue O, calcofluor white, or immunofluorescent stains via monoclonal antibodies) is very helpful to visualize cysts or trophozoite forms and therefore to differentiate infection from colonization, but this was not done in our center due to the estimated risk of aerosolization during COVID-19. As we have now a better knowledge on this viral pandemic, the use of these staining methods might be discussed, at least in some cases, using all necessary and extra precautions, in particular using a class II bio safety. This might be decisive in order to properly diagnose these co-infections. Serum (1,3)-*β*-D-glucan assay might be used for its negative predictive value to rule out the diagnosis of infection [7]. Finally, the diagnosis of infection must therefore be based, in addition to the mycological criteria, on a set of arguments including clinical worsening, immunosuppression, deep lymphocytopenia, serum (1,3)-*β*-D-glucan, and LDH assays and response to treatment [10].

The decision to treat or not treat these patients is also a matter of debate. In the series from Blaize *et al.*, the two patients improved without specific treatment, which was in favor of the diagnosis of colonization by *P. jirovecii* [6]. In the study from Alanio *et al*., four of the 10 patients received TMP-SMX as prophylaxis and six were not treated, among whom four rapidly improved. Three patients died, including one treated and two non-treated patients [7]. In the present study, all patients with COVID-19 and a positive *P. jirovecii* PCR were treated with curative dosage. For three of them (patients 1, 2, and 4), the decision was based on the presence of underlying diseases and mechanical ventilation, clinical worsening, positive serum (1,3)-*β*-D-glucan (in one patient), elevated LDH (in two patients), and compatible chest CT scan, which were suggestive of PCP infection. The last patient without mechanical ventilation had proven PCP diagnosed on the cytology from BAL fluid. The high mortality of the patients from the current series might encourage curative—at least prophylactic—treatment in severe patients with COVID-19. Finally, the decision to introduce curative treatment, prophylaxis, or no treatment is challenging and should take into account all the arguments described before to distinguish colonization and infection, but larger studies are mandatory to conclude on this question.

Our retrospective case-series study has some limitations. First, only 57 samples from COVID-19 patients were tested for *P. jirovecii* superinfection, and some cases were possibly missed. On the contrary, the diagnosis of PCP in COVID-19 is difficult, as mentioned before, especially since direct examination was not performed in COVID-19 patients, and some patients were possibly excessively treated only based on the single positivity of the *P. jirovecii* PCR. Finally, the prognosis of the patients included in this study was very poor, but whether this high mortality is due to pneumocystosis or if the occurrence of pneumocystosis is merely a marker of very severe COVID-19 and immunosuppression remains unknown.

Our study provides new data regarding *P. jirovecii* infections in patients with COVID-19. These co-infections are rare but severe, therefore, PCP should be considered in case of worsening of the condition of patients with severe COVID-19. Additional studies are needed to clarify the epidemiology of *P. jirovecii* in COVID-19 patients, the diagnosis, and the treatments.

## Figures and Tables

**Figure 1 jof-07-00585-f001:**
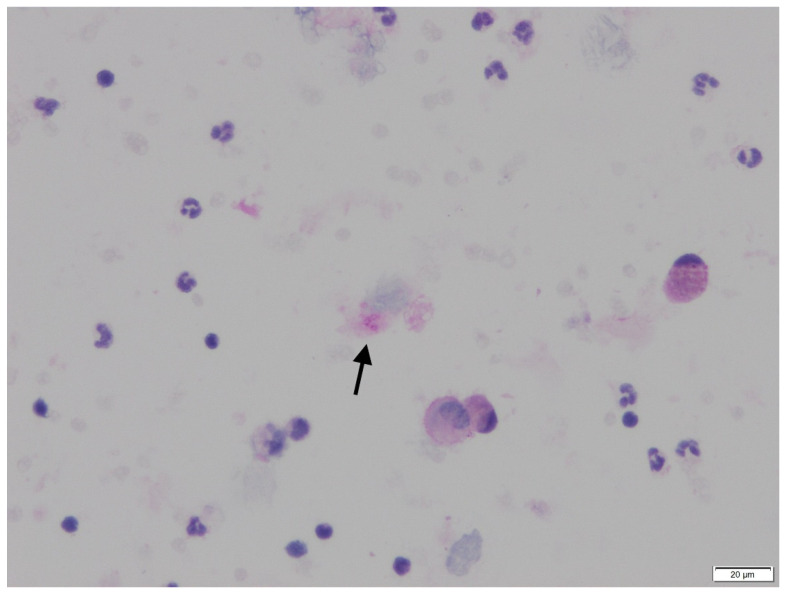
PAS (Periodic Acid Schiff) staining highlights the round cysts with a small central condensation (arrow) corresponding to the organisms of *Pneumocystis*, isolated on BAL cytology of patient 3.

**Table 1 jof-07-00585-t001:** Patients characteristics.

	Patient 1	Patient 2	Patient 3	Patient 4
Sex	F	M	M	M
Age	80	70	83	80
BMI	36	35	28	28
Underlying disease	Rheumatoid arthritis	Renal cell carcinoma in 2008, Autologous HSCT in 2013 for IgA myeloma complicated by amylosis, Ewing sarcoma with pulmonary metastases ongoing treatment, chronic kidney disease, pulmonary pneumocystosis six month ago	Diabetes mellitus, chronic heart failure, chronic kidney disease, atrial fibrillation, asthma	Diabetes mellitus, pacemaker for atrioventricular block, hypertension, transitory ischemic attack
Immunosuppressive treatment	Prednisone, Celecoxib	Cyclophosphamide, Doxorubicine	No	No
Pneumocystosis prophylaxis	No	TMP-SMX	No	No
Microbiological diagnosis				
COVID-19	PCR on NPS	PCR on NPS	PCR on NPS	PCR on NPS
*P. jirovecii*	PCR on BAL	PCR on tracheal aspirate	PCR and cytology on BAL	PCR on tracheal aspirate
Time between hospitalization and *P. jirovecii* PCR (days)	1	1	40	25
BDG (Fungitell Cape Cod.)	NR	NR	123 pg/mL	137 pg/mL
Biological data				
Lymphocytes count (cells/mmc)	300	222	570	600
LDH (IU/L)	NR	811	437	267
Clinical data				
Time from symptom onset to hospitalization	5 days	5 days	5 days	7 days
CT-scann	GGO >75%; AC	GGO > 75%; AC; nodules	GGO 50%; PE; crazy paving	GGO > 50%; AC; PE
ICU transfer *	Yes, day 1	Yes, day 1	No	Yes, day 30
Ventilation	Mechanical	Mechanical	Low flow nasal oxygen	HFNC then mechanical
Steroid therapy	Yes	No	Yes	Yes
Pneumocystosis treatment	Yes, TMP-SMX	Yes, TMP-SMX	Yes, TMP-SMX	Yes, TMP-SMX, then atovaquone
Outcome	Death 18 days after admission	Death 3 days after admission	Death 44 days after admission	Death 44 days after admission

F: female; M: male; BMI: body mass index; HSCT: hematopoietic stem cell transplantation; TMP-SFX: trimethoprime-sulfamethoxazole; PCR: polymerase chain reaction; NPS: nasopharyngeal swab; *P. jirovecii*: *Pneumocystis jirovecii*; BAL: bronchoalveolar lavage; BDG: serum (1,3)-β-D-glucan; NR: not recorded; LDH: lactate dehydrogenase; CT-scan: computed tomography scanner; GGO: Ground-glass opacities; AC: alveolar condensation; PE: pleural effusion; ICU: intensive care unit; HFNC: high-flow nasal oxygen with cannula. * ICU transfer after hospitalization.

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
