# Peer review of "Detection of Pneumocystis jirovecii in Patients with Severe COVID-19: Diagnostic and Therapeutic Challenges"

_jof, 2021, doi:10.3390/jof7080585_

Round 1

Reviewer 1 Report

Data reported citing reference 7 are wrong. In line 49, the article reports different numbers (the reference reports 10/108 cases with positive PCR). In line 108 the reference does not report the hypothesis that they were all colonizations. In line 123, the reference actually says “Whether a positive result should be an indication to consider administering co-trimoxazole, at least at prophylactic dosage in COVID-19 patients remains questionable” which is a bit different from what the authors write.

I would add CD4 and CD8 values in table 1, and BDG values as well, and the expand a bit the discussion including clinical reasoning on why patients were treated, considering the BDG values, eventual repeat tests, and a discussion on colonization vs infection. 

Line 132: maybe specify that BDG has value because of its negative predictive value, and as thus should be used to exclude the diagnosis, not confirm it. 

Line 140: expand on the clinical history of patients 3, since he was not in the ICU. Why did he die? Did you consider PJP to be the the other patients' main cause of death? 

General comments: The article is not very original, since bigger case series from the same country have already been reported. I still find it useful since so very little is known about Covid-PJP coinfection. I would however expand the discussion on colonization vs infection, and on the use of BDG to aid the diagnosis. 

Author Response

Dear Editor,

Please find enclosed a revised version of our manuscript, named “Detection of Pneumocystis jirovecii in patients with severe COVID-19: diagnostic and therapeutic challenges”

We thank the reviewer 1 for the constructive comments and suggestions to improve the quality of our manuscript. We have changed the content of the manuscript accordingly. We address below each of the issues raised by the reviewers in a point-by-point reply. We hope that the modified manuscript will be considered fit for publication.

All authors agree with this new version of our manuscript.

1) Data reported citing reference 7 are wrong. In line 49, the article reports different numbers (the reference reports 10/108 cases with positive PCR).

Response: Thank you for this comment. We used the data from the preprint version, which had been updated. We changed it accordingly.

2) In line 108 the reference does not report the hypothesis that they were all colonizations.

Response: We agree with this comment and modify our sentence.

“In the study from Alanio et al., four of the 10 patients received TMP-SMX as prophylaxis and six were not treated, among whom four rapidly improved. Three patients died, including one treated and two non-treated patients [7].”

3) In line 123, the reference actually says “Whether a positive result should be an indication to consider administering co-trimoxazole, at least at prophylactic dosage in COVID-19 patients remains questionable” which is a bit different from what the authors write.

Response: We also agree with this comment. This was based again on the preprint version. We have modified the paragraph on therapeutics.

“The decision to treat or not these patients is also debating. In the series from Blaize et al., the two patients improved without specific treatment, which was in favor of the diagnosis of colonization by P. jirovecii [6]. In the study from Alanio et al., four of the 10 patients received TMP-SMX as prophylaxis and six were not treated, among whom four rapidly improved. Three patients died, including one treated and two non-treated patients [7]. In the present study, all patients with COVID-19 and a positive P. jirovecii PCR have been treated with curative dosage. For three of them (patients 1,2 and 4), the decision was based on the presence of underlying diseases and mechanical ventilation, clinical worsening, positive serum (1,3)-β-D-glucan (in one patient) and elevated LDH (in two patients) and compatible chest CT scan, which were suggestive of PCP infection. The last patient without mechanical ventilation, had proven PCP diagnosed on the cytology from BAL fluid. The high mortality of the patients from the current series might encourage curative - at least prophylactic - treatment in severe patients with COVID-19. Finally, the decision to introduce curative treatment, prophylaxis or no treatment is challenging and should take into account all the arguments described before to distinguish colonization and infection, but larger studies are mandatory to conclude on this question.”

4) I would add CD4 and CD8 values in table 1, and BDG values as well and the expand a bit the discussion including clinical reasoning on why patients were treated, considering the BDG values, eventual repeat tests, and a discussion on colonization vs infection.

Response :

We have now added the BDG values in table 1 (P3 123 and P4 137 pg/mL Fungitell Cape Cod.). Unfortunately, the CD4 and CD8 values were not available.

We have added new paragraphs on the discussion between colonization and infection and on the therapeutic decision (lines 124-153).

5) Line 132: maybe specify that BDG has value because of its negative predictive value, and as thus should be used to exclude the diagnosis, not confirm it

Response: We thank the reviewer for this comment. We have modified by : “Serum (1,3)-β-D-glucan assay might be use for its negative predictive value to rule out the diagnosis of infection”

6) Line 140: expand on the clinical history of patients 3, since he was not in the ICU. Why did he die? Did you consider PJP to be the other patients' main cause of death? 

Three of them died in the ICU. The last patient (patient 3), died suddenly four days after TMP-SMX introduction for proven PCP in rehabilitation service; possibly attributed to PCP. The diagnosis was made by positive P. jirovecii PCR and cytology from BAL (figure 1) after pulmonary worsening 45 days after hospital admission.

We have added these new information lines 99-102.

Reviewer 2 Report

Dear Editor,

The article describes the presence of Pneumocystis jirovecci (diagnosed by PCR) in four out of 57 patients severely suffering from COVID-19 and with several co-morbidities.

The manuscript is well written, in good English and presents the major points of the relevant literature, the described situation and the limits that exist.

However as documented even by the authors the diagnosis of PCP is very challenging, especially in critically ill patients and mostly in COVID-19 where several clinical and laboratory similarities could exist between the two types of infection.

Only one of the four patients had a proven diagnosis according to the EORTC/MSG criteria and indeed; solely the positive PCR result without the presence of a staining methodology could just indicate a colonisation and not a real infection by Pneumocystis jirovecii.

Thus, although the main concept is that the scientific community must be very cautious with COVID-19 patients and enforce a rigorous searching for possible co-existence of PCP, the described cases do not really justify the manuscript’s title towards the conclusion of high mortality in COVID-19 probably due to the presence of  Pneumocystis jirovecii.

Probably the title should be something like: “Detection of Pneumocystis jirovecii in critically ill patients with COVID-19: diagnostic and therapeutic challenges” and therefore to analyse in the manuscript the dilemmas and the possible proposals starting with the authors’ case series and expanding it in the existing literature.

Sincerely yours.

Author Response

Response to the reviewer’s 2 comments:

Dear Editor,

Please find enclosed a revised version of our manuscript, named “Detection of Pneumocystis jirovecii in patients with severe COVID-19: diagnostic and therapeutic challenges”

We thank the reviewer 2 for the constructive comments and suggestions to improve the quality of our manuscript. We have changed the content of the manuscript accordingly. We address below each of the issues raised by the reviewers in a point-by-point reply. We hope that the modified manuscript will be considered fit for publication.

All authors agree with this new version of our manuscript.

We modified the title as suggested and expand the paragraph on diagnostic and therapeutic challenges.

We have modified by: “The diagnosis of PCP in COVID-19 patients and the distinction between infection and colonization are very challenging [6]. Patients hospitalized in the ICU for COVID-19 might be at risk of PCP due to mechanical ventilation, use of corticosteroid therapy, or the existence of a cytokine storm leading to marked alveolar-interstitial pulmonary tissue [1]. But there are radiological similarities between the two infections, the presence of cysts or fine reticular changes on CT-scan being in favor of pneumocystosis, but not constant and sometimes difficult to see [3,5,10]. The high sensitivity of PCR can lead to over diagnosis of P. jirovecii infection in colonized patients and the distinction between colonization and PCP can be difficult, especially in immunocompetent patients [5–7,10]. Direct examination is usually not performed in COVID-19 patients. Serum (1,3)-β-D-glucan assay might be use for its negative predictive value to rule out the diagnosis of infection [7]. Finally, the diagnosis of infection must therefore be based, in addition of the mycological criteria, on a set of arguments including clinical worsening, immunosuppression, deep lymphocytopenia, serum (1,3)-β-D-glucan and LDH assays and response to treatment [10].

The decision to treat or not these patients is also debating. In the series from Blaize et al., the two patients improved without specific treatment, which was in favor of the diagnosis of colonization by P. jirovecii [6]. In the study from Alanio et al., four of the 10 patients received TMP-SMX as prophylaxis and six were not treated, among whom four rapidly improved. Three patients died, including one treated and two non-treated patients [7]. In the present study, all patients with COVID-19 and a positive P. jirovecii PCR have been treated with curative dosage. For three of them (patients 1,2 and 4), the decision was based on the presence of underlying diseases and mechanical ventilation, clinical worsening, positive serum (1,3)-β-D-glucan (in one patient) and elevated LDH (in two patients) and compatible chest CT scan, which were suggestive of PCP infection. The last patient without mechanical ventilation, had proven PCP diagnosed on the cytology from BAL fluid. The high mortality of the patients from the current series might encourage curative - at least prophylactic - treatment in severe patients with COVID-19. Finally, the decision to introduce curative treatment, prophylaxis or no treatment is challenging and should take into account all the arguments described before to distinguish colonization and infection, but larger studies are mandatory to conclude on this question.”

Round 2

Reviewer 2 Report

Dear Editor,

I think that the manuscript is ok at its present form and would like to thank the authors for their modifications.

However and according to our experience as well, with great numbers of COVID patients investigated for possible fungal co-infections (and for PCP), a staining methodology should be attempted even in these cases. Of course all the necessary and extra precautions (class II bio safety, etc.) should be taken. It is always a very helpful and clearly diagnostic approach and according to our experience it hasn’t increased our laboratory’s risks.

Probably a word of advice in the text for the readership in order to start thinking on how to apply again the helpful staining methods would not be a bad idea.

Sincerely yours,

Author Response

We thank the reviewer 2 for the constructive comments and suggestions to improve the quality of our manuscript. We have changed the content of the manuscript accordingly. We address below each of the issues raised by the reviewers in a point-by-point reply. We hope that the modified manuscript will be considered fit for publication.

All authors agree with this new version of our manuscript.

1) Probably a word of advice in the text for the readership in order to start thinking on how to apply again the helpful staining methods would not be a bad idea.

Response: We agree with this comment and modify the text.

In results we suppress the following sentence “Unfortunately, direct examination and standard staining were not performed in the context of COVID-19 because of the risk of aerosolization.”.

In the discussion we add the following sentence: “The use of a staining methodology (Giemsa, Gomori-methenamine-silver stain, toluidine blue O, calcofluor white or immunofluorescent stains via monoclonal antibodies) is very helpful to visualize cysts or trophozoite forms and therefore to differentiate infection from colonization, but this has not been done in our center due to the estimated risk of aerosolization during the COVID-19. As we have now a better knowledge on this viral pandemic, the use of these staining methods might be discussed, at least in some cases, using all the necessary and extra precautions, in particular using a class II bio safety. This might be decisive in order to properly diagnose these co-infections. ».
